# Producing Iron Endohedral Fullerene on Electron Cyclotron Resonance Ion Source

**Yushi Kato** [1,*], **Takayuki Omori** [1], **Issei Owada** [1], **Wataru Kubo** [1], **Shuhei Harisaki** [1], **Koichi Sato** [1], **Kazuki Tsuda** [1], **Takumu Maenaka** [1], **Masahiro Anan** [1], **Masayuki Muramatsu** [2], **Atsushi Kitagawa** [2] and **Yoshikazu Yoshida** [3]

1. Division of Electrical, Electronic and Infocommunications Engineering, Graduate School of Engineering, Osaka University, 2-1, Yamada-oka, Suita-shi, Osaka 565-0871, Japan; omori@nf.eie.eng.osaka-u.ac.jp (T.O.); owada@nf.eie.eng.osaka-u.ac.jp (I.O.); w.kubo@nf.eie.eng.osaka-u.ac.jp (W.K.); harisaki@nf.eie.eng.osaka-u.ac.jp (S.H.); k.sato@nf.eie.eng.osaka-u.ac.jp (K.S.); k.tsuda@nf.eie.eng.osaka-u.ac.jp (K.T.); maenaka@nf.eie.eng.osaka-u.ac.jp (T.M.); anan@nf.eie.eng.osaka-u.ac.jp (M.A.)
2. National Institute of Radiological Science (NIRS), National Institutes of Quantum and Radiological Science and Technology (QST), 4-9-1, Anagawa, Inage-ku, Chiba 263-8555, Japan; muramatsu.masayuki@qst.go.jp (M.M.); kitagawa.atsushi@qst.go.jp (A.K.)
3. Faculty of Science and Engineering, Toyo University, 2100, Kugirai, Kawagoe-shi 350-8585, Japan; yyoshida@toyo.jp
* Correspondence: kato@eei.eng.osaka-u.ac.jp; Tel.: +81-6-6105-5913

**Abstract:** An electron cyclotron resonance (ECR) ion source (ECRIS) can generate an available amount of multicharged ions, thus it is not limited for use in the field of accelerator science, but also in medical/biological fields, such as for heavy ion beam cancer treatment and ion engines. The processes of generating multicharged ions are mainly sequential collisions of a direct ionization process by electrons, and have good ion confinement characteristics. By utilizing this confinement property, we have synthesized iron-encapsulated fullerenes, which are supramolecular and can be expected to have various high functions. Fullerenes and iron ions are vaporized from pure solid materials and introduced into the ECRIS together with the support gas. We investigated conditions under which fullerene ions do not dissociate and iron ions are generated so that both can coexist. Generated ions are extracted from the ECRIS and separated by mass/charge with a dipole magnet, and detected with a Faraday cup. This measurement system is characterized by a wide dynamic range. The charge-state distribution (CSD) of ion currents was measured to investigate the optimum conditions for supramolecular synthesis. As a result, a significant spectrum suggesting the possibility of iron-encapsulated fullerenes was obtained. This paper describes the details of these experimental results.

**Keywords:** ECRIS; ECR plasma; ion beams; multicharged ions; fullerene ions; iron ions; synthesized iron-encapsulated fullerenes

## 1. Introduction

Electron cyclotron resonance (ECR) ion sources (ECRIS) were diverted from fusion research to the field of accelerator science by Geller et al. in the early days [1]. Since then, ECRIS has been used for many years in related fields while achieving dramatic developments. Because its application is large enough to contribute to various uses, it is applied not only to the field of accelerator science but also to various fields, and is now used for heavy ion beam cancer treatment [2], ion engines in the aerospace field [3], bionano fields [4], and semiconductor fields.

In the 2000s, research and development of a miniaturized and advanced medical use-type of heavy ion beam cancer treatment using carbon multicharged ion beams was started, and fullerenes attracted attention as an allotrope as a carbon source that could be vaporized at a relatively low temperature as a carbon source. On the other hand, supramolecules

with new functions have been used to attract attention by including various elements in the inner space of fullerenes themselves. Endofullerenes are also relevant to fullerene astrochemistry [5]. Pioneering research had been started in the Hungarian JAERI group (ATOMK) [6], and at the same time, the Tohoku University group had also conducted nitrogen encapsulation experiments inside fullerenes by depositing a substrate in plasma using a grid based on a Q-machine [7].

Toyo University, NIRS, Hungary JAERI(ATOMK), and Osaka University groups started joint research on the establishment of a new bio-nano ECRIS at Toyo University, and in the 2010s, encapsulation experiments inside fullerenes were conducted by the ion implantation method and by using a grid inside the ECRIS [8,9].

From the summer of 2014, we started the endohedral fullerene vapor phase synthesis experiment at ECRIS (Osaka University, Osaka, Japan). A tandem-type ECRIS device was constructed [10], and first, in a fullerene ion beam generation experiment, multicharged fullerene ions up to the triple-charged level and beam extraction were successful [11]. Then, we started an experiment on nitrogen-encapsulating fullerene formation, and succeeded in obtaining a spectrum suggesting nitrogen-encapsulating fullerene formation from the analysis of the mass/charge distribution of the extracted ion beam current [12].

Fullerene ion beams in ECRIS can be generated with extremely low power of several watts by introducing fullerene vapor under high vacuum. However, fullerenes dissociate at high microwave power under normal ECRIS operating conditions for multicharged ion generation. We have succeeded in the stable generation of a fullerene ion beam by applying a telecommunication device with an output of several watts as a microwave source [13].

On the other hand, in ECRIS, a certain amount of microwave power is used to produce an iron ion beam. Therefore, it is difficult to select ECRIS operating conditions that match fullerene ion generation. The Osaka University group introduced iron vapor using an induction heating (IH)-type iron evaporation source, and succeeded in experimentally finding the operating conditions of ECRIS in which the fullerene ion beam and the iron ion beam coexist. It has become possible to measure the charge-state distribution (CSD) of the beam current under the condition that both coexist [14]. Therefore, in the experiment from January to March 2020, the measurement of the CSD of the ion beam current in the same experiment led to the acquisition of experimental spectral data, suggesting the formation of iron-encapsulated fullerene. Moreover, in July 2020, an ion beam from ECRIS in the absence of iron vapor was acquired, and there was no spectrum corresponding to the mass number of iron-encapsulated fullerenes, confirming the above-mentioned encapsulation fullerene formation.

In addition, in order to cross-check the formation of iron-encapsulated fullerenes, the mass spectra corresponding to fullerenes, iron, and iron-encapsulated fullerenes were also found from the analysis results by TOFMA, etc. of the samples obtained with the toluene solvent of the deposits near the electrodes in the ECRIS [14]. At present, we are trying to acquire more accurate data while the continuous experiment is interrupted by the new coronavirus syndrome.

## 2. Experimental Apparatus

Figure 1 shows a top view of the entire ECRIS device (Graduate School of Engineering, Osaka University) used in the gas phase synthesis experiment of iron-encapsulated fullerenes. The fullerene evaporation source is introduced from the downstream side of the extraction electrode side of the ECRIS vacuum vessel, and the IH-type iron evaporation source is inserted from the upstream mirror end along the vacuum vessel axis.

The magnetic field configuration constructed by a pair of mirror coils A and B, and an auxiliary coil C for adjustment near the resonance point formed in the center of the mirror magnetic field are superposed with an octupole magnetic field by a permanent magnet [10]. The distribution of the magnetic field strength along the mirror axis with respect to a typical coil current is shown in Figure 1 below. When generating a normal multicharged ion beam, the same amount of current is applied to coils A and B to form

a symmetric mirror magnetic field for operation. On the other hand, when introducing fullerene vapor from the side of the ECRIS vacuum vessel, the optimum condition for fullerene ion beam generation found experimentally is to reduce the coil B current and extend the ECR resonance point to the downstream extraction electrode side. It has been found that the magnetic field coordination to face for the fullerene evaporation source to ECR zone is suitable for their ion beams.

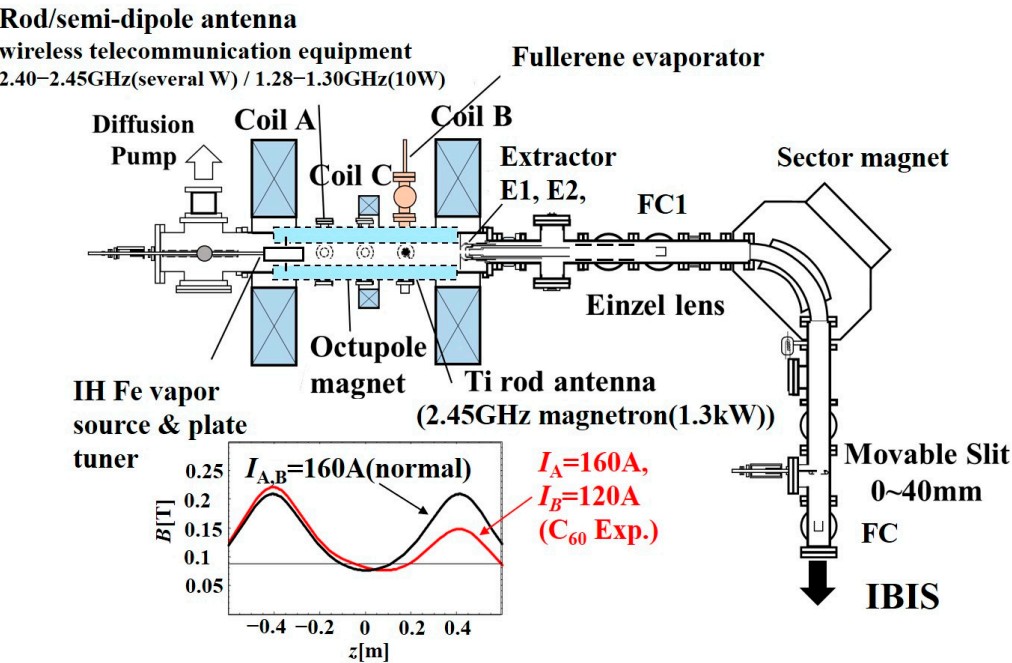

**Figure 1.** ECRIS (Osaka University) schematic drawing.

The ECRIS vacuum vessel is mainly exhausted by an oil diffusion pump of 3000 L/s, and the beamline near the extraction electrode is exhausted by two turbo molecular pumps, and the ultimate vacuum are $10^{-5}$ and $10^{-6}$ Pa range, respectively. Various gases are introduced by mass flow controller (MFC) or variable leak valve, and the operating pressure is about $10^{-4}$ to $10^{-3}$ Pa. In fullerene and iron-encapsulated fullerene synthesis experiments, high-purity rare gases such as He, Ar, and Xe are mainly used as support gases. In addition, high-purity $N_2$ gas was used in the nitrogen-encapsulated fullerene formation experiment.

2.45 GHz microwaves for ECR discharge are coaxially converted from the waveguide to the inside of the vacuum vessel via a three-dimensional circuit from a magnetron source with a maximum power of 1.3 kW, and inside the vacuum vessel through the waveguide coaxial conversion window. It is introduced with a rod antenna made of Ti. On the other hand, since a very low power microwave power supply is sufficient for fullerene ion beam generation, a microwave power supply for radio equipment was used. There are two types of frequency ranges, which can be adjusted in the bands of 2.40 to 2.45 GHz and 1.27 to 130 GHz, respectively. The former can supply 0.1 to several watts with a rod antenna, and the latter can supply several watts to several tens of watts with a semi-dipole antenna.

The generated ions are extracted from the extraction electrodes PE, E1, and E2 installed at the end downstream of the mirror magnetic field to form a beam. The normal extraction voltage is $V_{PE}$ = 10 kV, $V_{E2}$ = 0 (GND), but in the case of a fullerene ion beam, the mass number is large and $V_{PE}$ = 1 to 2 kV due to the upper limit of the magnetic field strength of the analysis dipole electromagnet. The extracted ion beam is mass-charge-separated by a sector electromagnet, which is a dipole electromagnet for analysis, and then the resolution and beam current amount are adjusted by a movable slit and collected by a Faraday cup (FC). The current is measured using a log amplifier or a linear amplifier. Normally, the extracted voltage is kept constant, and the magnetic field of the sector electromagnet is scanned to measure the charge-state distribution (CSD) of the extracted ion beam currents.

Since the dynamic range of this CSD measurement system is wide, ranging from mA to pA, the vapor phase synthesis of iron-encapsulated fullerenes in this study is similar to that of secondary ion mass spectrometry (SIMS) used for detecting dopants in semiconductors. It is considered to be advantageous for detection compared to various chemical analyses based on sampling.

On the downstream side, after FC, the ion beam irradiation system (IBIS), which conducts beam measurement such as emittance and beam distribution and irradiation, experiments on various materials, especially artificial satellite constituent materials, after decelerating to low energy. It can be operated independently from the beam line [15]. It is also possible to use this IBIS to decelerate and deposit, resulting in a thin film from the ion beam.

Figure 2 shows the fullerene evaporation source and the induction heating (IH)-type iron evaporation source used in the iron-encapsulated fullerene experiment. The fullerene evaporation source is a crucible made of $Al_2O_3$; a 0.5 mmf Ta wire is used for the heater, and several tens of watts are supplied by a DC power supply for heating. The IH-type iron evaporation source consists of a double induction heating coil made by Mo coils, and the IH power supply for consumer use is supplied as a power supply with a maximum of 6.4 kW with the insulated induction heating coil transformer (IHCT), which we have developed originally. In this experiment, the maximum input power to the power supply was mainly about 600 to 800 W. Please refer to the past literature for the heating characteristics of each evaporation source [11,13,16].

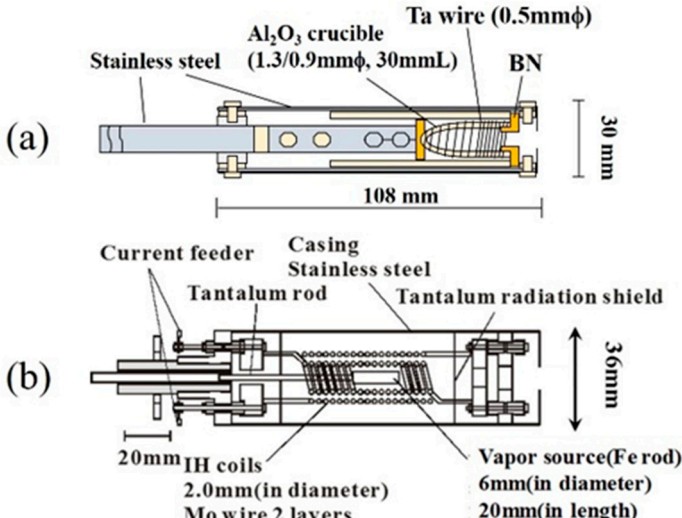

**Figure 2.** Schematic diagram of fullerene evaporation source (**a**) and induction heating (IH)-type iron evaporation source (**b**).

## 3. Experimental Results

In this study, after establishing both iron and fullerene ions generation conditions, we experimentally investigated the operating conditions that enable the coexistence of fullerene ions and iron ions with ECR plasma. Then, in the charge-state distribution (CSD) of the ion beam current extracted under the operating conditions, the detailed spectrum corresponding to the mass/charge of the iron-encapsulated fullerene and its identification were performed. The scheme for finding the experimental conditions of ECRIS that enables these coexistences is explained, and the target spectrum is shown as below.

### 3.1. Procedure to Make $Fe^+$ and $C_{60}^+$ Coexist on ECRIS

Since the formation conditions of iron ions and fullerene ions in the ECR plasma are different, the conditions under which they coexist were investigated. The fullerene evaporation source is preheated without Ar support gas until just before the fullerene

vapor is emitted. After that, the input power of induction heating (IH) iron evaporation is gradually increased to raise iron vapor. After the iron ion beam reaches the temperature near the temperature where the detectable iron vapor is generated, the support gas is introduced and the ECRIS device is started. Then, the ion beam is extracted at the extraction voltage $V_{PE}$ = 10 kV, and the temperature of the IH iron evaporation source is gradually raised while checking the iron ions. The operating and extraction conditions of the ion source are optimized so that the $Fe^+$ ion beam current is maximized under the constraints of relatively low microwave power and low operating pressure.

From here, while checking the $Fe^+$ ion beam, we gradually approach the production conditions suitable for the fullerene ion beams. First, the extraction voltage $V_{PE}$ is gradually lowered from 10 kV to 1–2 kV, that is the extraction voltage at which fullerene ions can be analyzed. At that time, the coil A, B, and C currents, microwave power, plate tuner, and ion beam extraction conditions ($V_{E1}$, gap between PE-E1 electrodes, etc.) are optimized so that the spectrum of the $Fe^+$ ion beam can be checked and constantly observed. Next, the temperatures of the fullerene evaporation source and the IH iron evaporation source are gradually raised to optimize the fullerene ion and iron ion beams, and while maintaining their coexistence, the optimum conditions for the mirror magnetic field and microwave power. Additionally, the operation of the ECRIS gradually shifts to conditions suitable for generating fullerene ion beams while maintaining the iron beam current [13].

Finally, the spectrum corresponding to the endohedral fullerene ion beam is confirmed by detailed observation on the high mass side of the fullerene beam. Then, this last step is repeated to find the optimal conditions for gas phase synthesis of endohedral fullerenes in ECRIS.

### 3.2. Detail Spectrum Focusing Nearby and Heavier Side of $C_{60}^{q+}$

After the equipment of the ECRIS was relocated and rebuilt from January to February 2016, the microwave source for radio equipment was applied to fullerene beam generation and the IH iron evaporation source was optimized from December 2016 to October 2017. From December 2017, we started the gas phase synthesis experiments of iron-encapsulated fullerenes in ECRIS itself, and from January to March 2018, the spectrum in which iron ions and fullerenes coexist in ECRIS was obtained by the procedure described in the previous section. However, during that time, only a few experiments focusing on the formation of iron-encapsulated fullerene ions could be performed. We were planning to continue this experiment, but we had to spend about one and a half years after the fall of 2019 in recovery after the earthquake in the northern part of Osaka in June 2018.

We started preparations for this experiment after the fall of 2019, and conducted this experiment again from January to March 2020, after a hiatus of two years. The procedure is as described in the previous section, but in the final stage of exploring the generation of iron-encapsulated fullerenes, a more accurate linear amplifier was used from the log amplifier. In the experiment, during about three months, there were about three opportunities to investigate the iron-encapsulated fullerene spectrum, and after detailed identification, we came to confirm the spectrum that is the target repeatedly and with good reproducibility. The representative data obtained are shown below.

Figure 3 shows the data obtained using a log amplifier for the typical coexistence spectrum of iron and fullerene ions obtained during this experiment. The support gas is Ar, and Xe gas remains for other experiments. The extraction voltage $V_{PE}$ is 2 kV. Microwave power is about 10 W. The input power to the iron evaporation source and the fullerene evaporation source is about 600 W and about 11 W, respectively.

Figure 4 shows the data acquired by a linear amplifier with a typical spectrum on the high mass side of the $C_{60}^+$ and $C_{60}^{2+}$ spectra. Fullerene ions have been observed up to triple-charged ions including their fragments, and $Fe^+$ and $Xe^+$ (including isotopes) have also been clearly observed. In this figure, the detailed spectra obtained by increasing the amplifier amplification factor are shown on the right side of $C_{60}^+$ and $C_{60}^{2+}$. Detailed identification shows the spectrum corresponding to the mass number indicating the inclusion of Ar, Fe,

and Xe, etc. Regarding single charges Xe inclusions, it is considered that the inclusion spectrum was observed because both $C_{60}^+$ and $C_{58}^+$ overlap with the $C_{70}^+$ spectrum, and because $C_{56}^+$ is the largest in the dissociation spectrum of $C_{60}^+$. Furthermore, regarding inclusions of Ar and Fe, corresponding spectra in the right side near the $C_{60}^{2+}$ spectrum were also observed. Since it is detected as a beam, from the viewpoint of stability, it is highly possible that these are contained rather than replaced or adhered to C, which constitutes the $C_{60}$ ion. Figure 5a,b show enlarged figures on the high mass side near $C_{60}^{2+}$ and $C_{60}^+$ in Figure 4, respectively. The spectra of these results have been observed with good reproducibility in the three experiments this time.

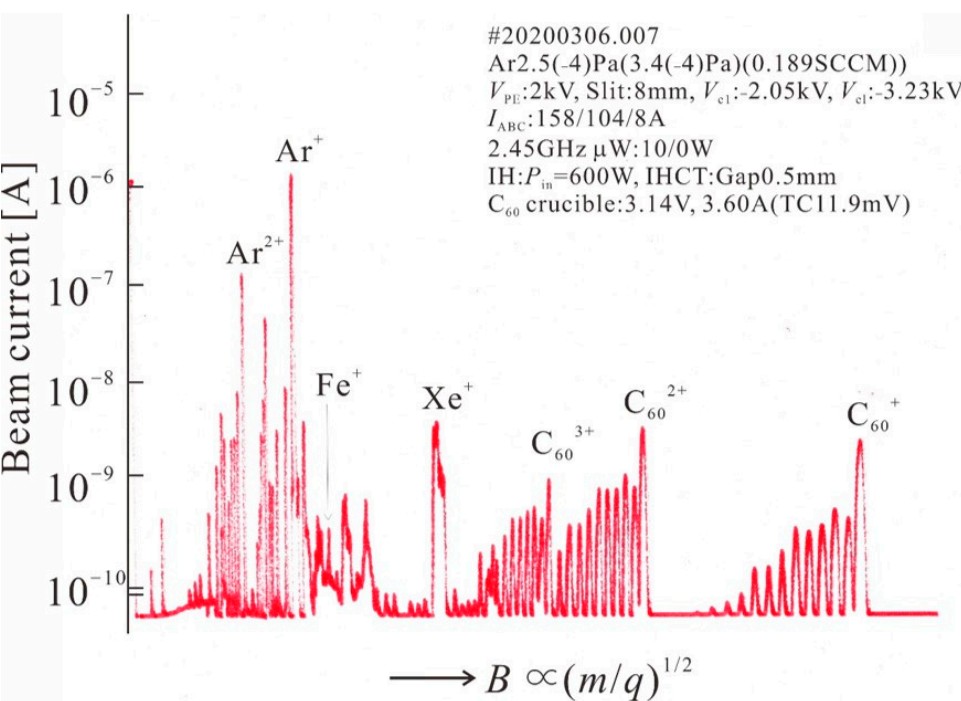

**Figure 3.** Typical coexistence spectrum of iron and fullerene ion (log amplifier).

Regarding the process related to formation mechanism of iron-encapsulated fullerene in ECRIS plasma at vapor phase, the formation mechanism is as follows. Electromagnetic waves are introduced into ECRIS with strong magnetic field under ultra-high vacuum, causing electron cyclotron resonance and generating plasma. Since it is a volume-generated plasma, it has a spatial potential of several tens of eV, and it is thought that the iron ions ionized in this plasma collision with $C_{60}$ with that energy, iron-endohedral fullerenes are generated, and they are contained and extracted to form a beam. It is considered that high stability is required without dissociation in ECRIS plasma, and, moreover, in order to be transported and analyzed as the beams.

Figure 6 shows the spectrum obtained when only fullerenes were introduced, and iron vapor was not introduced from the IH iron evaporation source under the coronavirus pandemic in July 2020 (Figure 6a,b). This is compared with the spectrum obtained in Figure 5 above (Figure 6c). Since the spectrum corresponding to the iron-encapsulated fullerene was not obtained without the introduction of iron vapor, it is considered that the reliability of the above-mentioned iron-encapsulated fullerene formation was improved.

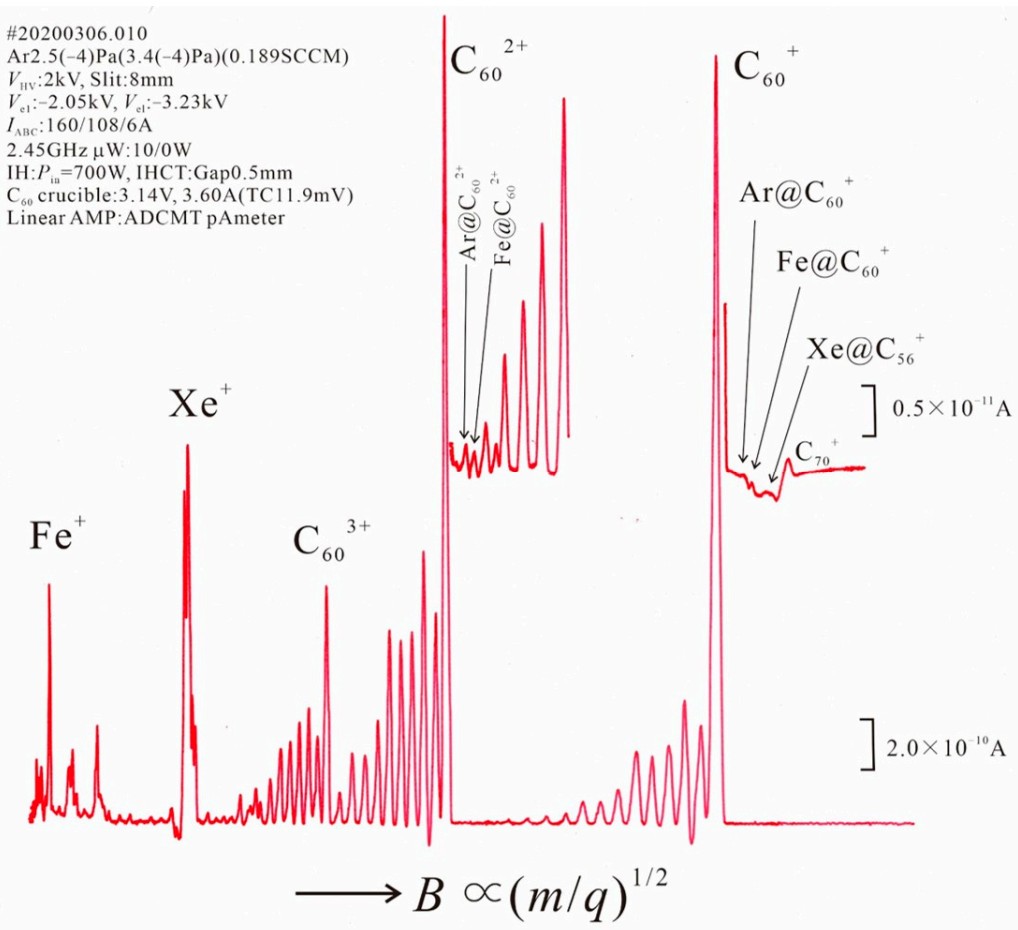

**Figure 4.** Typical spectrum on the high mass side of the $C_{60}^+$ and $C_{60}^{2+}$ spectra (linear amplifier).

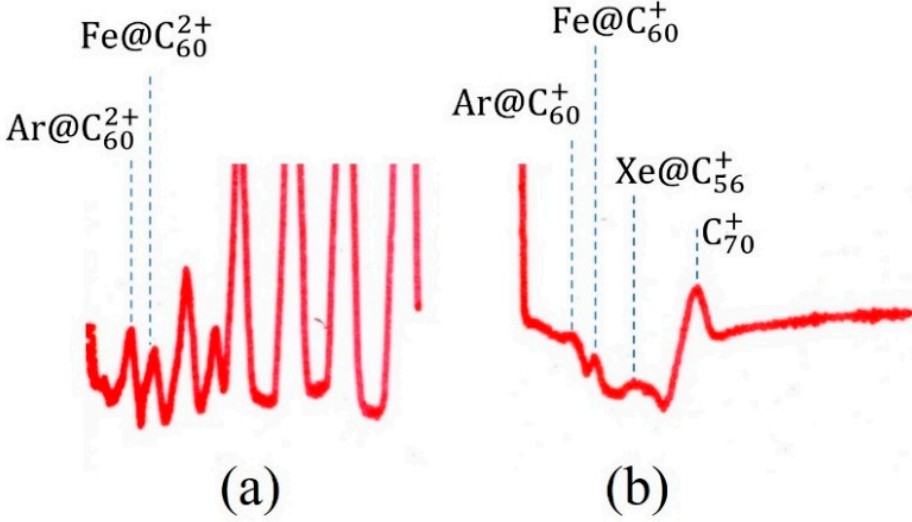

**Figure 5.** Detail spectrum on the high mass side of the $C_{60}^+$ and $C_{60}^{2+}$ spectra (linear amplifier).

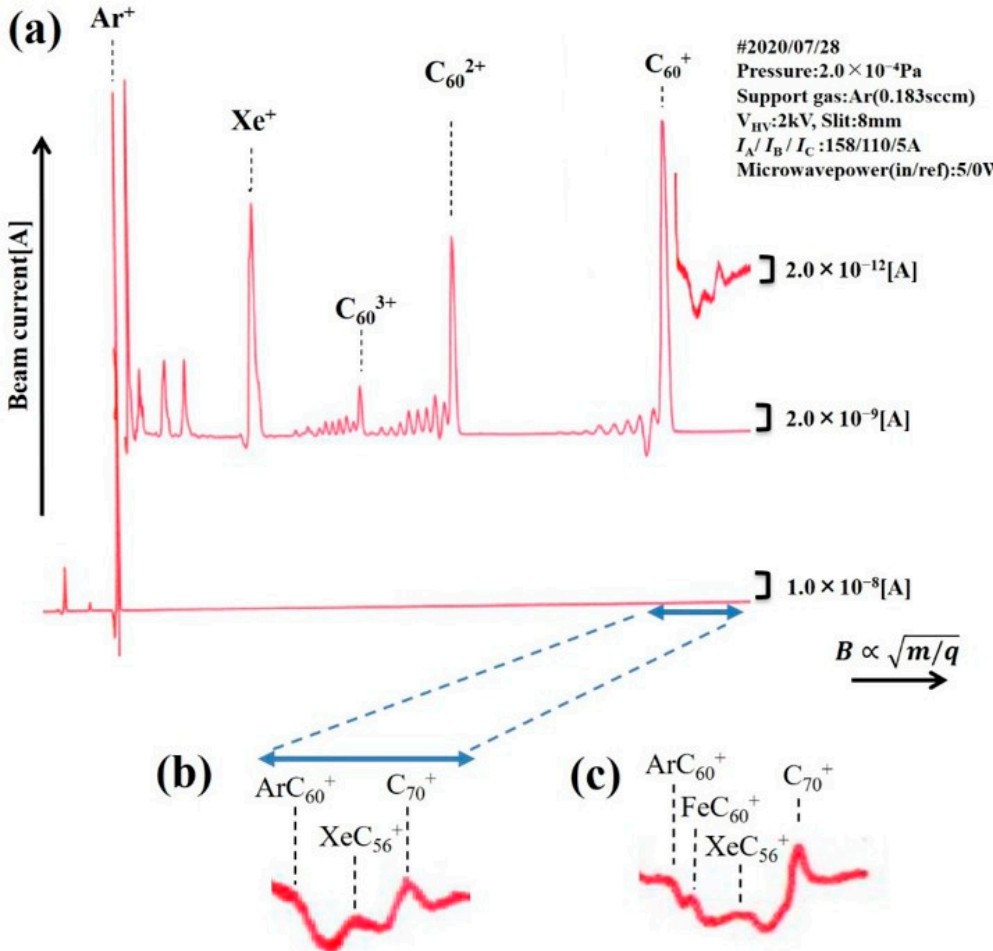

**Figure 6.** The spectrum obtained when only fullerene was introduced without iron vapor (**a**), and (**b**) for comparing that with ion vapor in Figure 4 (and Figure 5b) (**c**).

## 4. Discussion

In the above experiment, we focused on the mass charge analysis of the ion beam current extracted from the ECRIS. If it is detected from the viewpoint of stability, it is considered that there is a high possibility of contained fullerenes, but it is necessary to further increase the current value. For this purpose, it is necessary to increase the vapor fluxes of fullerenes and iron, respectively. Furthermore, cross-check analysis to support the formation of iron fullerene compounds and iron-encapsulated fullerenes is considered indispensable. We, the Osaka University group, have long used time-of-flight mass spectrometry (TOFMA) and high-performance liquid chromatography mass spectrometry (HPLCMA) to analyze the deposits adhering to the inner wall of the ECRIS vacuum vessel. Further analysis is needed and will be analyzed in the future.

## 5. Conclusions

ECRIS is currently being applied in various fields because it can generate a usable amount of multicharged ions. In this study, in order to further expand the field of application, we aimed to apply it to supramolecular vapor phase synthesis for a system with good confinement. As a target, we selected iron-encapsulated fullerenes, which have not yet been included in cases. For fullerenes, the crucible of resistance heating was used as the evaporation source, and for iron, the vapor of pure substances was introduced into ECRIS by the induction heating (IH) evaporation source. Since the production conditions of iron ions and fullerenes differ greatly depending on the ECRIS operating conditions, after experimentally investigating the conditions under which these can coexist, the charge-state

distribution (CSD) measurement of the ion beam generated in detail revealed that iron-encapsulated fullerenes were obtained. We were able to obtain well-reproducible spectra suggesting various inclusions. The formation of iron-encapsulated fullerenes is considered to be highly reliable because it is not observed in the spectrum without iron vapor. Since these results were extracted from ECRIS plasma, it is highly probable that iron is contained in the fullerenes, but future cross-checks are required to determine ultimately whether iron location is inside, on, or outside the $C_{60}$ fullerene cage. In addition, since there is room for improvement in the amount of fullerene evaporation and those of iron evaporation and furthermore improvement of the measurement system, we plan to apply these to experiments and analyses with the aim of increasing the iron-encapsulated fullerene beam current in the near future.

**Author Contributions:** Conceptualization, Y.K., Y.Y. and A.K.; methodology, Y.K., M.M., T.O. and W.K.; software, Y.K. and M.A.; validation, T.O., W.K. and I.O.; formal analysis, I.O. and T.M.; investigation, S.H.; resources, M.M.; data curation, I.O., T.M., K.S. and K.T.; writing—original draft preparation, Y.K.; writing—review and editing, Y.K.; visualization, Y.K.; supervision, Y.K.; project administration, Y.K.; funding acquisition, Y.Y. and A.K. All authors have read and agreed to the published version of the manuscript.

**Funding:** This research received no external funding.

**Institutional Review Board Statement:** Not applicable.

**Informed Consent Statement:** Not applicable.

**Acknowledgments:** The author provided informative discussion and continued encouragement by A. G. Drentje (KVI), S. Biri, R. Rácz (ATOMKI), and Takashi Uchida (Shin-Etsu Chemical). We would like to thank all the staff and graduates of Osaka University for their great efforts in preparing and constructing this experimental device. We would like to thank Hisanao Kazama and Kazumi Yano of Osaka University for the TOFMA measurement and experimental preparation.

**Conflicts of Interest:** The authors declare no conflict of interest.

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
