# Peer review of "Producing Iron Endohedral Fullerene on Electron Cyclotron Resonance Ion Source"

_crystals, doi:10.3390/cryst11101249_

Round 1

Reviewer 1 Report

This manuscript by Y. Kato et al. reports on the iron endohedral fullerene synthesis through the collision between C60 and iron ions by electron cyclotron resonance ion source technique. The products were separated by mass/charge with a dipole magnet and detected by a Faraday cup. The topic is highly appealing to the field of endohedral metallofullerene synthesis. Iron endohedral metallofullerene synthesis through the most widely used direct current arc discharge method appeared problematic, alternative synthesis approaches may bring surprising results on this topic. By checking with the reported results, I am not convinced by the claimed success with iron endohedral fullerene synthesis. I will be very happy to read the revision considering the following comments.

1, The reported FeC60+ signal is not convincing because the peak is too weak to determine reliably. The comparison between C60 ions with/without iron ions could be supplementary evidence, but not a decisive one in this case.

2, The authors proposed further experimental analysis on the products with HPLC-MS, it is appealing in this work. The results may clear the ambiguity of the current evidence.

3, The authors should comment on the iron endohedral fullerene with precautions even with much more solid evidence from the HPLC-MS analysis. Indeed, with the mass spectrum, it is still hard to determine if the iron is encapsulated in the fullerene cage. It could be attached outside the wall of fullerene (known as exohedral fullerene) or on the wall of fullerene (known as heterofullerene). It is known that iron is keen to coordinate with the aromatic carbon rings, such as the famous ferrocene.

Author Response

The author would like to thank the reviewers. The author's response to the comment is included in the attached cover letter. Please see it.

Reviewer 2 Report

The manuscript describes a production technique of iron endofullerene with cyclotron electron resonance. I cannot recommend it without revising due to following reasons:

(1) The identification of Fe@Cx is based only on the mass spectra. The authors should clearly point how they conclude about endohedral location of iron atom.

(2) The authors must strengthen the scientific soundness of the study in introduction. Please, mention that iron endofullerenes are relevant to fullerene astrochemistry (see and mention Omont's review article in Astronomy & Astrophysics [https://doi.org/10.1051/0004-6361/201527685 ]).

(3) It is known that doubly filled endohedral complexes of fullerene can be produced including the noble gas endofullerenes mentioned in the work. Why the doubly filled species were not observed? Please, discuss this issue in the context of Sabirov et al. work, RSC Adv. 2016 [https://doi.org/10.1039/C6RA12228K] whereby it was shown that the formation of such endofullerenes under high-energy impacts is regulated with symmetry rules in addition to statistical laws.

(4) Please, make wider the discussion section. I suppose the author have something to say. For exapmle, the mechanistic considerations and possibly response to point (3) of this review.

In general, I find the work interesting in two aspects. In narrow one, the work contributes to fullerene chemistry and astrochemistry. In general aspect, it is important for understanding the behavior of matter under high-energy impacts.

Author Response

(The authors gave the same response as above.)

Round 2

Reviewer 1 Report

The authors clarified part of my concerns in the previous review report, however, not fully. The existence of FeC60+ is acceptable based on the authors’ argument. The endohedral nature of Fe in C60 is not convincing in my opinion. As indicated in the results, such as Figures 4 and 6, etc. the fragmentation of fullerene presents in the system, which means that fullerene structure transformation could happen in the system. So, the argument of Fe encapsulation is not so convincing. The Fe could be located on the cage or outside the cage. The authors’ argument on ferrocene is not exactly my point in the previous review report. I am sorry for the misleading caused by the unclear description. Ferrocene is attempted as an example of Fe located outside the fullerene cage, just like ferrocene structure (or half ferrocene), the Fe could be located outside the fullerene cage, coordinates with the pentagon/hexagon of the fullerene cage, just like the ferrocene structure. Since the experiment is in the gas phase, these structures could not be excluded by stability arguments. It is hard to determine the structure unambiguously at the current stage. I recommend the authors make some comments on the possibility of Fe location (inside/on/outside the C60 fullerene cage) in the manuscript to avoid misleading.

Author Response

The authors would like to thank Reviewer # 1 for his constructive and respectable comments.

Reviewer 2 Report

I can’t say that the authors convinced the reviewers in the endohedral location of iron metal inside the fullerene cage. Of course, the complexes surviving under extreme conditions of this work should be endohedral. Nevertheless, this statement is better justified with experiment. 
 The another are asked to add a sentence about their assumption (strongly evidenced) about endohedral structure of the complexes. Other point to improve deals with the formal authors’ attitude to the discussion section (even after revision)

Author Response

The authors would like to thank Reviewer # 2 for his constructive and respectable comments.
